# Multicow pose estimation based on keypoint extraction

**Caili Gong[1,2,3], Yong Zhang[1]\*, Yongfeng Wei[2,3], Xinyu Du[2], Lide Su[1], Zhi Weng[1,2,3]**

**1** College of Mechanical and Electrical Engineering, Inner Mongolia Agricultural University, Hohhot, China, **2** School of Electronic Information Engineering, Inner Mongolia University, Hohhot, China, **3** State Key Laboratory of Reproductive Regulation and Breeding of Grassland Livestock, Inner Mongolia University, Hohhot, China

\* yongz@imau.edu.cn

**Data Availability Statement:** The minimal data set underlying the results described in our paper can be found at https://www.kaggle.com/twisdu/dairy-cow. All data in our paper will be fully shared without restriction.

## Abstract

Automatic estimation of the poses of dairy cows over a long period can provide relevant information regarding their status and well-being in precision farming. Due to appearance similarity, cow pose estimation is challenging. To monitor the health of dairy cows in actual farm environments, a multicow pose estimation algorithm was proposed in this study. First, a monitoring system was established at a dairy cow breeding site, and 175 surveillance videos of 10 different cows were used as raw data to construct object detection and pose estimation data sets. To achieve the detection of multiple cows, the You Only Look Once (YOLO)v4 model based on CSPDarkNet53 was built and fine-tuned to output the bounding box for further pose estimation. On the test set of 400 images including single and multiple cows throughout the whole day, the average precision (AP) reached 94.58%. Second, the keypoint heatmaps and part affinity field (PAF) were extracted to match the keypoints of the same cow based on the real-time multiperson 2D pose detection model. To verify the performance of the algorithm, 200 single-object images and 200 dual-object images with occlusions were tested under different light conditions. The test results showed that the AP of leg keypoints was the highest, reaching 91.6%, regardless of day or night and single cows or double cows. This was followed by the AP values of the back, neck and head, sequentially. The AP of single cow pose estimation was 85% during the day and 78.1% at night, compared to double cows with occlusion, for which the values were 74.3% and 71.6%, respectively. The keypoint detection rate decreased when the occlusion was severe. However, in actual cow breeding sites, cows are seldom strongly occluded. Finally, a pose classification network was built to estimate the three typical poses (standing, walking and lying) of cows based on the extracted cow skeleton in the bounding box, achieving precision of 91.67%, 92.97% and 99.23%, respectively. The results showed that the algorithm proposed in this study exhibited a relatively high detection rate. Therefore, the proposed method can provide a theoretical reference for animal pose estimation in large-scale precision livestock farming.

**Funding:** This study was funded by the National Natural Science Foundation of China under Grant 61966026, Grant 62161034 and Grant 61561037. The funders had no role in study design, data collection and analysis, decision to publish, or preparation of the manuscript.

**Competing interests:** The authors have declared that no competing interests exist.

## Introduction

The external behavior of dairy cows is a comprehensive reflection of their well-being and conditions. Daily poses (standing, walking, lying) can reflect the activity level of cows because cows generally reduce activity and increase lying during illness and show mounting behavior during estrus. It is very time-consuming, costly and subjective to record the individual information of dairy cows by long-term manual observation. Currently, many researchers have used various sensors to detect the behavior of dairy cows [1–4]. Wearable sensors pose certain disadvantages that may cause stress responses in dairy cows. Due to the advantages of long-term and noncontact continuous monitoring, machine vision has been used to monitor livestock activity and health in precision livestock farming. Recently, a large number of studies on the behavior detection of dairy cows based on machine vision have been conducted, such as lameness detection [5–7], estrus detection [8, 9] and prediction of the time of calving [10–13]. Studies have found that changes in the pose of cows can reflect their health and provide important data support for lameness detection, estrus detection, and prediction of calving.

Currently, research on human pose estimation is relatively advanced and can accurately realize pose estimation in complex backgrounds. The bottom-up method using keypoint heatmaps was used for human pose estimation with a small model and high efficiency [14–17]. Li et al. [18] proposed a top-down method to tackle the problem of pose estimation in the crowd; this method used a single human pose detector to identify humans and then detected key points on each human frame.

In recent years, animal pose estimation has received increasing attention, and much research has been performed on the adoption of deep learning algorithms for this task. Talmo et al. [19, 20] proposed a LEAP framework for tracking body-part positions of fruit flies under controlled light and uniform background and tested the method on freely moving laboratory mice. Liu et al. [21] introduced a video-based animal pose estimation architecture that took into account variability in animal body shape and temporal context from nearby video frames. This method exhibited high precision when tested on datasets of mice, zebrafish, and monkeys. Hahn Klimroth et al. [22] presented a multistep convolutional neural network for detecting three typical poses of African ungulates, obtaining a high accuracy of 93%. Zheng et al. [23] introduced Faster R-CNN on a deep learning framework to identify five poses (standing, sitting, sternal recumbency, ventral recumbency and lateral recumbency) and obtained accurate sow locations in loose pens. The estimation of pose change can reveal the health of the sow. Chen et al. [24] proposed an algorithm based on YOLACT with high detection speed and accuracy for real-time detection and tracking of multiple parts of pig bodies. Based on the prediction of keypoints, Song et al. [6] proposed a skeleton extraction model of cows in walking states with a high accuracy rate of up to 93.40% when the OKS was 0.75. Since the color of breeding sites is similar to the body color, pose estimation of dairy cows is more difficult than that for other animals. Relatively few studies have been performed to estimate the pose of cows.

Based on long-term manual observation, the daily poses of dairy cows mainly include standing, walking, lying and transitioning between standing and lying. Therefore, we developed a daily pose estimation algorithm for multiple cows in an actual farm environment and performed experiments on the proposed model. We mainly focused on the classification of three daily poses (standing, walking, lying). The main scheme of pose estimation is to detect the object in each frame and then classify the different poses in the image. First, the YOLO v4 [25] model was built and fine-tuned to output the accurate object frame. Then, we extracted the keypoint heatmaps and PAFs of cows and matched the keypoints with the same cow based on a human body keypoint estimation model. Finally, the cow skeletons in different poses

were input into the classification model to estimate the three typical poses of cows (standing, walking and lying).

## Materials and methods

This study was carried out at Inner Mongolia Flag Animal Husbandry Co., Ltd. in Inner Mongolia Autonomous Region of China. Inner Mongolia Agricultural University has conducted scientific research with Inner Mongolia Flag Animal Husbandry Co., Ltd. for more than five years. The study does not require approval from the relevant authorities. There are no ethical issues. The data was acquired by the monitoring camera, which was fixed on the fence of the breeding site at a height of 4 m. During the experiment, neither the data acquisition equipment nor the experimental personnel contacted the cows and had no stress response to cows. Compared with traditional manual inspection and using wearable sensors, it can realize non-contact animal behavior detection and improve animal welfare.

### Data acquisition

All data in this study was acquired at Inner Mongolia Flag Animal Husbandry Co., Ltd. between September 2017 and April 2018. Ten cows were housed in this barn with an activity area that was 35 m long and 20 m wide, ensuring sufficient space for free movement. To ensure the continuous around-the-clock monitoring of the cows' behavior, two cameras with 5 million pixels infrared mode (Hikvision Inc., Zhejiang, China) were mounted at the fence of the breeding site at a height of 4 m. The surveillance video format was MP4. The image resolution and frame rate were 1920*1080 pixels and 25 fps, respectively. The videos were transmitted to a net video recorder with a 1T hard disk through the network cable and then copied to a computer through the USB interface every week for further processing. The background of the videos was the field of activity, and the cows walked freely without handlers.

### Image labeling and data processing

**(1) Data set of object detection.** In this study, surveillance videos were used as the raw data, and each video was approximately 25 minutes long. Images were selected from 175 videos of 10 cows. To increase the robustness of the object detection model, images with single and multiple objects with occlusion were selected. Since the data used in this study were acquired throughout the day, the illumination conditions in the collected images also differed according to the movement of the sun. LabelImg, an open-source image labeling tool, was used to manually annotate the training set. The area of the annotation rectangle box was as small as possible to reduce the unrelated background pixels. We manually annotated the bounding boxes for 1800 sampled images for object detection that were divided into a training set (1620 images) and a validation set (180 images) at a ratio of 9:1. To improve the robustness of the detection algorithm, a mosaic augmentation algorithm was used to enrich the training set. It randomly selected 4 pictures from the total data set, performed mirroring, flipping, rotation, cropping and stitching at random positions to synthesize new images. After mosaic augmentation, the total number of images in the training set was 3240. A separate data set (400 images) was then selected as the test set from the surveillance video of 10 cows.

**(2) Data set of keypoint detection and pose classification.** According to the daily poses of cows in the activity field, there are three main types of poses: standing, walking and lying. Single cows with different poses were manually selected from the surveillance video, so that the images contain each condition (standing, walking, lying). The open-source tool Labelme [26] was used to manually label the keypoints of cows in 1800 images, including 600 images each for standing, walking and lying, which were divided into a training set and validation set

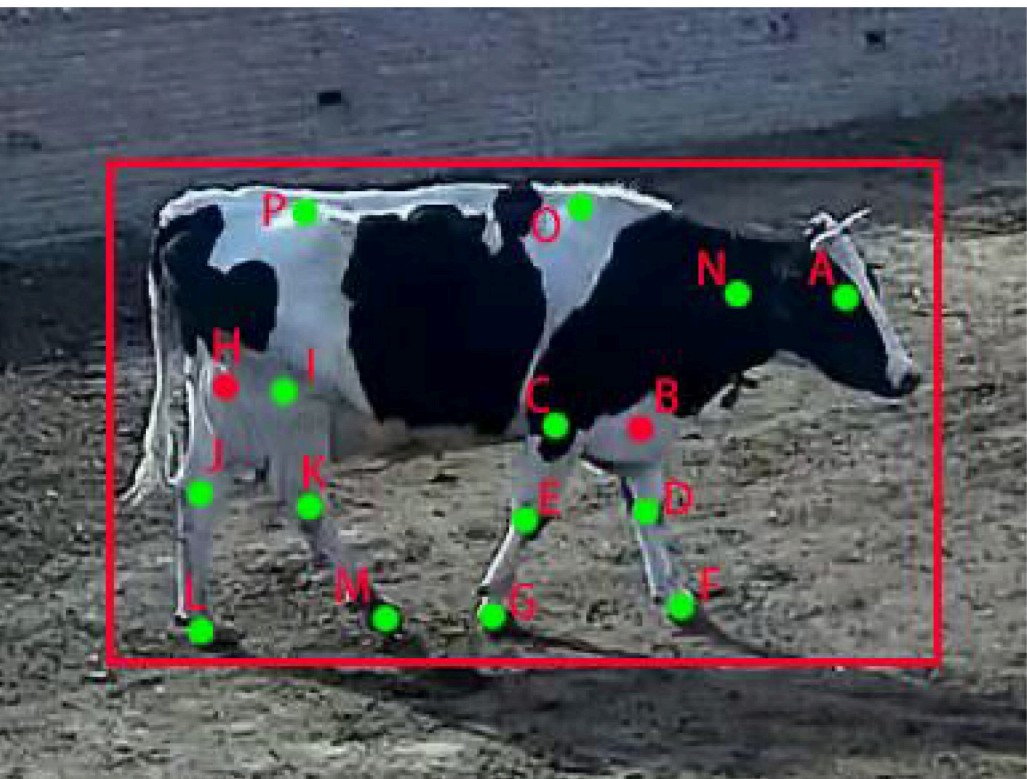

**Fig 1. Keypoint label template of the cow skeleton.**

at a ratio of 9:1. A sample image of keypoint labels is shown in Fig 1. These 16 keypoints (marked A, B, . . ., P) represent the head, left front leg root, right front leg root, left front knee, right front knee, left front hoof, right front hoof, left hind leg root, right hind leg root, left hind knee, right hind knee, left hind hoof, right hind hoof, neck, spine and coccyx. The visible keypoints (marked green) are named as 2, and the invisible point (marked red) is named as 1. The missing keypoints (marked on the top left corner of the image) were named 0. After obtaining the keypoint information under different behaviors, the pose data set was marked. Considering the influence of light conditions and occlusion on pose estimation, 390 images (including 130 images each for standing, walking and lying poses) with single cow and double cows with occlusion under different light conditions were used as the test set.

## Methods

The methods used in the previous studies of pose estimation are mainly divided into two types: the top-down methods that first detected objects prior to body part estimation, and bottom-up methods that first detected body parts and then grouped them into objects. In the top-down framework, the network for multianimal pose estimation mainly contains LEAP [19, 20], DeepPoseKit [27] and DeepLabCut [28], in which the detected individual objects are first captured and analyzed separately. The accuracy of pose estimation depends on the result of object detection, which is prone to missed detection and incomplete interception of the whole cow image. The bottom-up frame method is mainly used for multiperson pose detection. To improve the accuracy of pose estimation, we combine object detection with a pose estimation

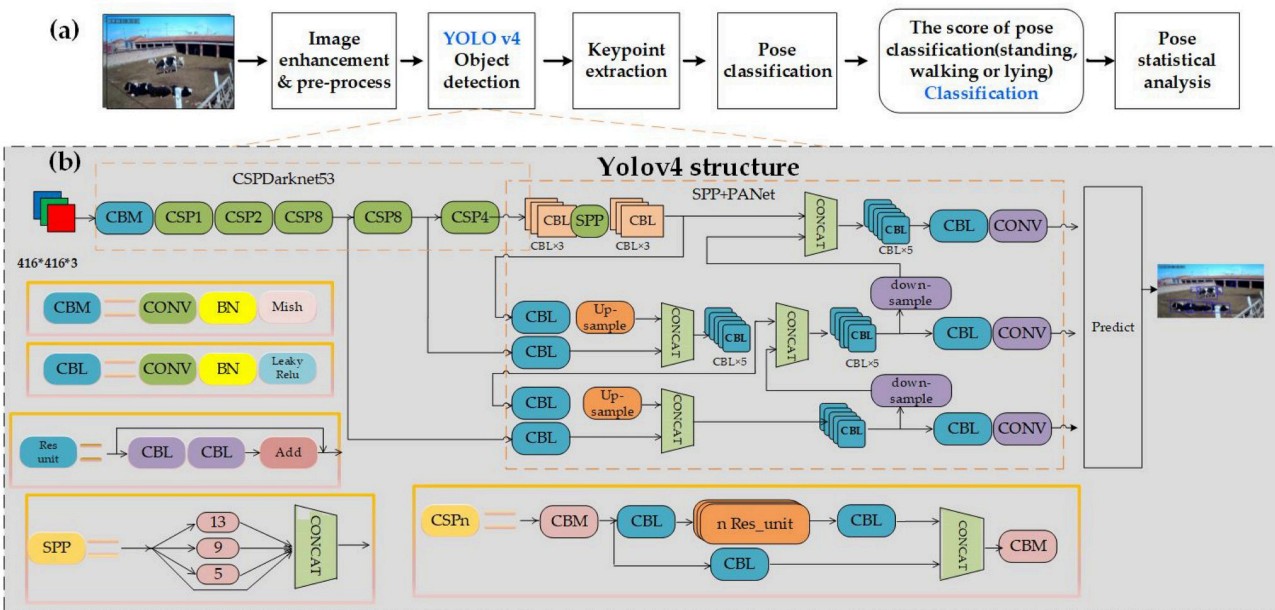

**Fig 2.** (a) Overview of pose analysis procedure. (b) Close-up of the multiobject detector based on YOLO v4.

network in this study. Fig 2. shows the flow of the automatic analysis outputting the classification and confidence score from the pose estimation network.

**Multiobject detection.** The results for the separation between the cows and background directly affect the accuracy of pose estimation. To achieve the initial separation of the object and background and output the bounding box of cows for pose estimation, we built a multiobject model prior to pose estimation. In recent years, much research has been conducted on multiobject detection [29–31]. In this study, the YOLO v4 network which is a one-stage object detection algorithm was used to carry out fast and accurate detection of multiple cows.

As shown in Fig 2(b), the CSPDarkNet53 neural network framework was used as the backbone to extract image features. The backbone contains multiple residual blocks that consist of convolution, batch normalization, and a Mish activation function. PANet (path aggregation network) integrated the extracted features to improve the detection performance.

Three feature layers were obtained by the backbone for detecting small, medium, and large objects that were fused through PANet. Therefore, each image was divided into $S^*S$ grids (S was 52, 26, and 13) that were used to detect small, medium and large objects. Each grid cell was responsible for checking the object bounding box position, confidence, and type of the center point within it. The confidence reflects whether the current grid cell contains the cow objects and the corresponding prediction accuracy. The calculation formula can be expressed as follows:

$$\begin{cases} x_{\min} = \sigma(t_x + C_x - \dfrac{t_w}{2}) \\ y_{\min} = \sigma(t_y + C_y - \dfrac{t_h}{2}) \\ x_{\max} = \sigma(t_x + C_x + \dfrac{t_w}{2}) \\ y_{\max} = \sigma(t_y + C_y + \dfrac{t_h}{2}) \end{cases} \qquad (1)$$

where $(C_x, C_y)$ are the coordinates of the current grid cell, $(t_x, t_y)$ are the horizontal and vertical offsets of the center point, and $(t_w, t_h)$ are the width and height of the object detection box.

The original loss function in YOLO v4 consists of three parts, namely the bounding box regression loss, the confidence loss and the classification loss. In this paper, we modified the overall loss function into confidence loss and regression loss. Confidence loss was used to describe whether there was an object in the grid cell by calculating the binary cross entropy. Regression loss was used to describe the position and size difference between the annotated object and the predict object. Considering the three most important factors (overlapping area, center point distance and aspect ratio), the regression loss uses CIoU instead of IoU, as shown in Eqs (2)–(4).

$$L_{CIoU} = 1 - IoU + \frac{d^2}{c^2} + \alpha\upsilon \tag{2}$$

$$\upsilon = \frac{4}{\pi^2}\left(\arctan\frac{\omega^{gt}}{h^{gt}} - \arctan\frac{\omega}{h}\right)^2 \tag{3}$$

$$\alpha = \frac{\upsilon}{(1 - IoU) + \upsilon} \tag{4}$$

The network was pretrained on the PASCAL VOC 2012 dataset, and then the convolution layers were fine-tuned to achieve a better training effect using the self-built training dataset. According to GPU performance, the input image was set to 416 pixels * 416 pixels in training. The batch size and number of iterations were set as 2 and 50, and the initial learning rate and maximum learning rate were 0.000001 and 0.001, respectively. For the same object, the network may output multiple bounding boxes. We kept the bounding box with the highest confidence and excluded other boxes that had a high degree of overlap with it. Finally, we achieved the approximate separation of the object and background and output the accurate object frame to provide a reference location for dairy pose estimation.

**Skeleton extraction.** After achieving the approximate separation of object and background, the keypoint extraction network of dairy cows was constructed, and the pose classification network was built to realize the extraction of three typical poses (standing, walking and lying) [16, 17, 32]. The structure of the skeleton extraction model is shown in Fig 3.

*(1) Network structure.* The network structure consists of two branches: the upper branch is responsible for predicting the location of keypoints, and the lower branch is responsible for the partial affinity field between keypoints.

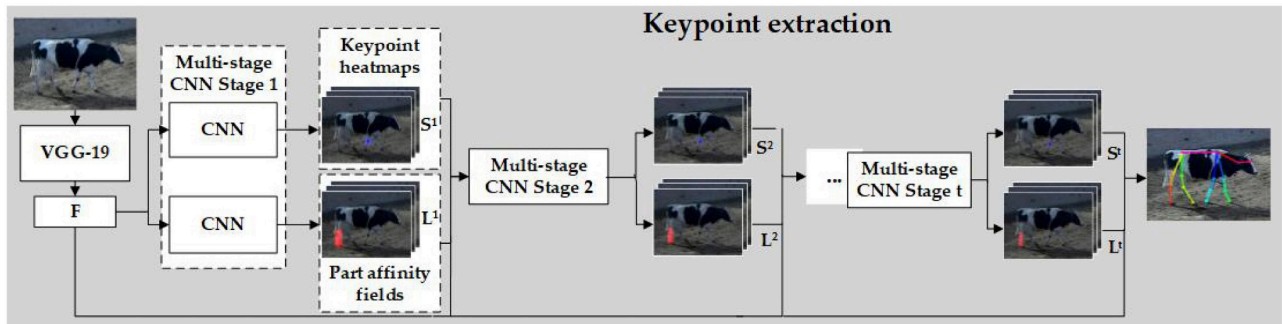

**Fig 3. The structure of two-branch skeleton extraction network.**

First, the feature maps F were extracted from the image by backbone VGG-19 (the first 10 layers without pooling). In most cases, the behavior will only appear in a small spatial area in the image. The low-level convolutional layer features contain more object location and detail information, and the high-level features have stronger semantic information. Therefore, we combined low-level features such as texture, color and edges with high-level features. The feature generated by the fourth convolution layer was downsampled, and the channel after the eighth layer was compressed. Then, the two feature maps were connected with the feature map output by the last convolution layer. The correlation information between different spatial areas is captured through feature fusion.

A multistage CNN (multistage convolutional neural network) was used to generate keypoint heatmaps and a part affinity field (PAF). $S(S_1, S_2, \ldots, S_J)$ represents the confidence map of keypoints that refers to the probability that keypoints appear in each pixel area. $L(L_1, L_2, \ldots, L_C)$ represents vector fields between every two keypoints that encodes the correlation degree among all keypoints. The first stage of the two-branch network consists of three convolution layers with a kernel size of $3^*3$ and two convolution layers with a kernel size of $1^*1$. The upper branch network predicts the keypoint confidence map $S^1 = \rho^1(F)$, and the lower branch network predicts part affinity field heatmap $L^1 = \varphi^1(F)$, where $\rho$ and $\varphi$ represent the CNN structure of the convolutional neural network. In each subsequent stage, both branches consist of five convolution layers with a kernel size of $7^*7$ and two convolution layers with a kernel size of $1^*1$. The prediction result of the previous stage merged with the original image feature is used as the input of the next stage. After multiple stages of operation, the keypoint prediction accuracy is improved. The network structure is expressed as follows.

$$S^t = \rho^t(F, S^{t-1}, L^{t-1}), \forall t \geqslant 2 \tag{5}$$

$$L^t = \varphi^t(F, S^{t-1}, L^{t-1}), \forall t \geqslant 2 \tag{6}$$

In the training phase, L2 loss is used to supervise the keypoints and part affinity field. In this study, the pretrained model of the COCO2017 person keypoint data set was used to initialize the network parameters. Then, the convolution layer was fine-tuned to achieve a better training effect on the self-built dataset.

*(2) Keypoint heatmap detection.* The heatmap represents the confidence that a keypoint appears in a certain position of the image that is composed of a series of two-dimensional points. We use a Gaussian kernel with fixed variance to determine the marked keypoint confidence of each position. For the jth keypoint of the kth person, $x_{j,k}$ is used to represent the actual position of the keypoint, and the confidence value of the pixels around the keypoint is expressed as:

$$S^*_{j,k}(p) = \exp\left(-\frac{\| p - x_{j,k} \|_2^2}{\sigma^2}\right) \tag{7}$$

where the standard deviation $\sigma$ controls the distribution range of the confidence value. For an image with multiple cows, the actual heatmap of each keypoint is the maximum value within the Gaussian kernel.

*(3) Part affinity field association.* The pixels on the limbs are represented by a unit vector, including the position and direction information, and all unit vectors on the limbs constitute the part affinity field. This is mathematically expressed by:

$$L^*_{c,k}(p) = \begin{cases} v & if \ p \ on \ the \ limb \ \mathrm{c} \\ 0 & otherwise \end{cases} \tag{8}$$

where v is the unit vector, and the part affinity field is defined as all unit vectors on the limb. If overlapping limbs of multiple cows k at point p are found, the field is the mean value of all vector fields.

The association confidence between any two keypoint positions $d_{j1}$ and $d_{j2}$ is obtained by linear integration of the part affinity field.

$$E = \int_{u=0}^{u=0} L_c(p(u)) \cdot \frac{d_{j2} - d_{j1}}{\parallel d_{j2} - d_{j1} \parallel_2} du \tag{9}$$

where $p(u)$ is the interpolation of the two positions.

*(4) Keypoint matchings with PAF.* Due to the uncertainty regarding the number of objects and occlusion in the image, there may be several candidates for each part. Greedy relaxation was used to generate optimal matches. The specific operations are as follows.

First, a point set of heatmaps of different cows is obtained to estabish a unique match between different point sets. The keypoints and PAF are regarded as the vertices and edge weight of the graph, respectively. Then, the multiobject detection problem is transformed into binary graph matching, and the optimal matching of the linked keypoints is obtained by using the Hungarian algorithm. Finally, the keypoints belonging to the same object in the image are marked on the object.

**Pose classification.** After extracting the position information of the keypoints of the cow's body in the image, pose estimation was performed based on the cow's skeleton in the object detection frame. Therefore, a fully connected neural network (FCNN) was designed to classify three typical behaviors (standing, walking and lying). The network structure is shown in Fig 4.

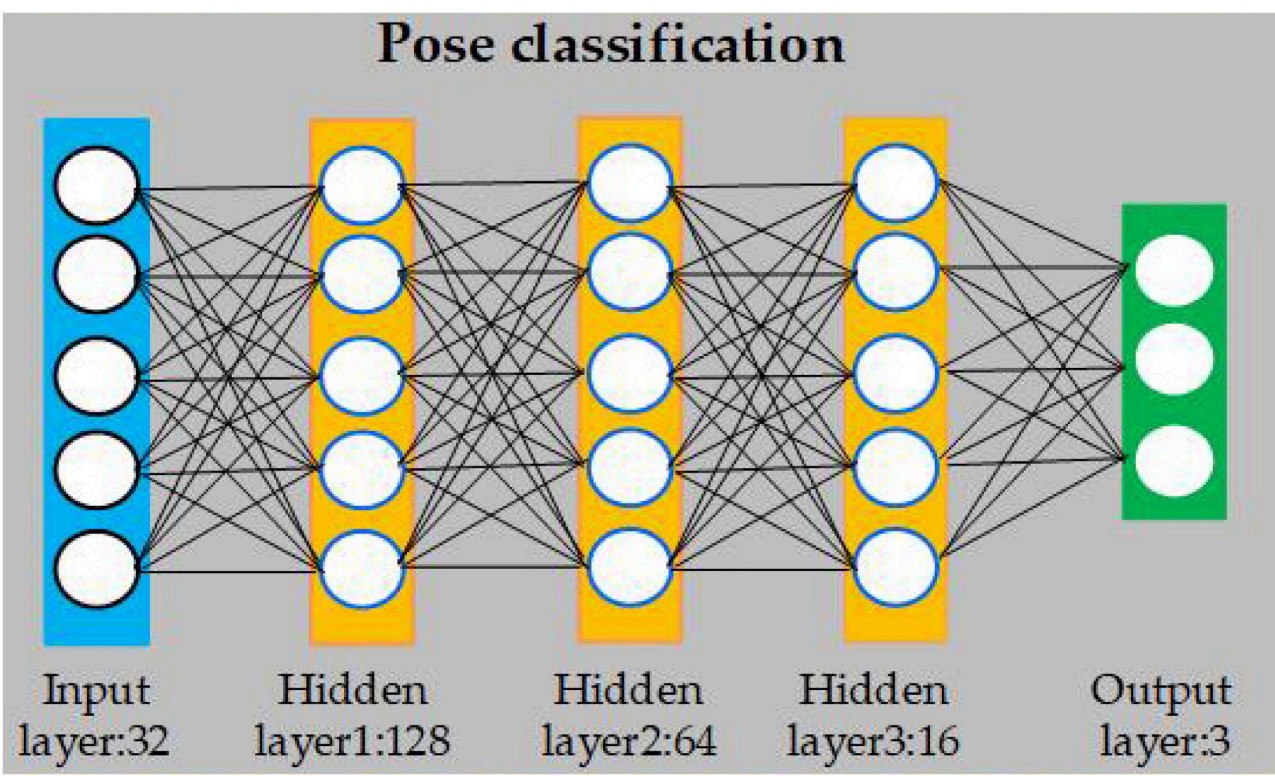

**Fig 4. Fully connected neural network of pose classification.**

Because there are 16 keypoints on the cow's body and each keypoint corresponds to 2 coordinates on the x-axis and y-axis, the input layer is set as 32 layers. The effective information is extracted and integrated through 3 hidden layers (128, 64, and 16). Each layer introduces the ReLU activation function, and the batch normalization layer is added to prevent the gradient from disappearing. The output layer is a dense layer with a size of 4 activated by softmax, and is used to predict the three typical behaviors. The softmax function maps the output to a value in the interval (0,1) that calculates the probability of each of the three behaviors.

## Results

All experiments were performed on a Windows 10 operating system with an Intel(R) Core (TM) with a 2.7 GHz CPU, 128 GB RAM and a 24 GB NVIDIA Quadro P6000 GPU. The model was written by using an open-source deep learning framework named TensorFlow 2.2 based on Python 3.7.

### Evaluation of multiobject detection

To verify the effectiveness of the algorithm, four indices namely precision, recall, average precision (AP) and detection speed were adopted in the evaluation of the object detection model. When IoU(Intersection over Union)is greater or equal to the threshold, the prediction result is considered to be a true positive case(TP). When IoU is lower than the threshold, it is considered to be false positive case (FP). When IoU is equal to 0, it is false negative case(FN). In this paper, the average precision(AP50 and AP75) was calculated when the IoU thresholds are set to 0.5 and 0.75, respectively. AP is the average value when an object is detected. The formulas for the calculation of precision, recall and AP are as follows:

$$\text{precision} = \frac{TP}{TP + FP} \tag{10}$$

$$\text{recall} = \frac{TP}{TP + FN} \tag{11}$$

$$AP = \int_0^1 p(r)dr \tag{12}$$

where TP, FP, FN and TN indicate the numbers of true positives, false positives, false negatives and true negatives, respectively.

The precision, recall and AP of this method are shown in Table 1. AP50 and AP75 represent the average precision when the IOU is equal to 0.5 and 0.75. Detection speed can reach 8.06 f/ s. It is observed from the test results that the detection accuracy of multiple cows was lower than that of a single cow because it was difficult to extract features when multiple cows were occluded. There was little difference in the detection accuracy between day and night. This was because the object detection dataset in this study was randomly selected from 24 hours of surveillance video. Therefore, the detection accuracy of the model remained nearly the same throughout the whole day. Generally, the multiobject detection method based on machine vision proposed in this study exhibited good accuracy.

### Evaluation of keypoint extraction model

To verify the performance of the algorithm, OKS (object keypoint similarity) was adopted as the evaluation index of the keypoint extraction model in this study. OKS plays the same role as IoU and is calculated between the predicted keypoints and ground truth keypoints. The

**Table 1. Evaluation indicators of object detection under different scenarios.**

| Test result | IOU | Precision(%) | Recall(%) | AP(%) | AP50(%) | AP75(%) |
|---|---|---|---|---|---|---|
| Single cow without occlusion | 0.5 | 99 | 99 | 78.07 | 99 | |
| | 0.75 | 98 | 98 | | | 97.61 |
| Multi-cows with occlusion | 0.5 | 99.28 | 95.8 | 71.25 | 95.8 | |
| | 0.75 | 94.2 | 90.91 | | | 88.75 |
| day | 0.5 | 99.55 | 98.67 | 72.03 | 98.67 | |
| | 0.75 | 94.64 | 93.81 | | | 91.98 |
| night | 0.5 | 96 | 84.85 | 60.27 | 82.77 | |
| | 0.75 | 89.14 | 78.79 | | | 74.4 |
| 400 images tested simultaneously | 0.5 | 98.56 | 94.49 | 69.4 | 94.04 | |
| | 0.75 | 93.59 | 89.72 | | | 87.07 |

calculation of OKS is shown in formula (13).

$$OKS = \sum_i \left[ \exp(-d_{p^i}^2 \Big/ 2s_{p^i}^2 k_i^2) \delta(v_{p^i} = 1) \right] \Big/ \sum_i [\delta(v_{p^i} = 1)] \qquad (13)$$

where d is the Euclidean distance between each ground truth and detected keypoint and p and i represent the IDs of object cows and keypoints, respectively. s is the object scale, and k represents the normalized factor of the ith keypoint. v is the visibility marker for the ground truth; $\delta$ is the Kronecker function.

We also take AP to account for the complete test images when the OKS threshold is 0.5. The calculation of AP follows the formula below.

$$AP = \frac{\sum_p \delta(OKS_p > 0.5)}{\sum_p 1} \qquad (14)$$

Considering the influences of light and occlusion on the results in this study, 390 images including single cows and dual cows with occlusions were tested under different light conditions. The experimental results are shown in Figs 5 and 6. Parts No. 1-16 represent the head, left front leg root, right front leg root, left front knee, right front knee, left front hoof, right front hoof, left hind leg root, right hind leg root, left hind knee, right hind knee, left hind hoof, right hind hoof, neck, spine and coccyx, respectively. Generally, the confidence of the 12 keypoints representing the legs was significantly higher than that of other keypoints. The average confidence of leg keypoints was 83.3% for single cows and 81.2% for double cows during daylight, and decreased slightly at night, with balue of 73.1% for single cows and 71.4% for double cows. The average values of the coccyx, spine, and neck sequentially followed the values for leg keypoints, and the head exhibited the lowest confidence.

It is observed from the experimental results that the detection precision of the leg keypoints was the highest, and was greater than 90% regardless of day or night and single cow or double cows. This was followed by the values for the back, neck, and head. For a single cow during the day, the AP of the coccyx was 83%, that of the spine was 65%, and those of the neck and head were slightly lower at approximately 61.5% and 51.5%, respectively. For double cows, the AP of the coccyx was 74.5%, that of the neck was 63.5%, and those of the spine and head were slightly lower at approximately 41.5% and 36.5%, respectively. At night, the AP values were lower for all keypoints for both single cows and double cows. The average detection accuracy of a single cow was 85% during daylight and 78.1% at night, and the average detection accuracies of double cows were 74.3% and 71.6%, respectively.

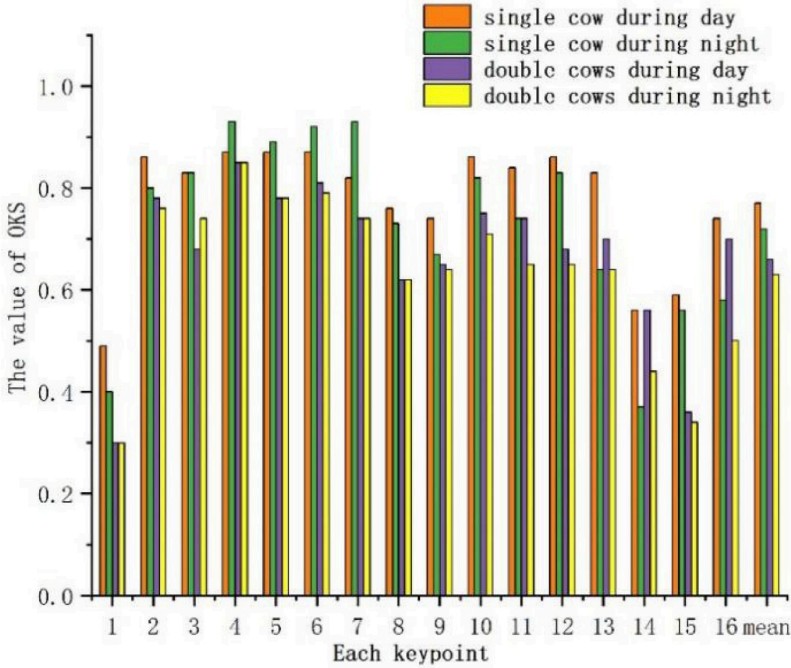

**Fig 5. The confidence of various keypoints in different scenarios.**

## Evaluation of pose classification model

In most multiclassification tasks, the evaluation metrics are computed based on the confusion matrix. In this study, the confusion matrix was also used as the evaluation metric of three typical pose classifications. Each column of the confusion matrix represents the predicted poses, the total number of which represents the number of predicted poses. Each row represents the true poses, the total number of which represents the actual amount of each pose. The larger

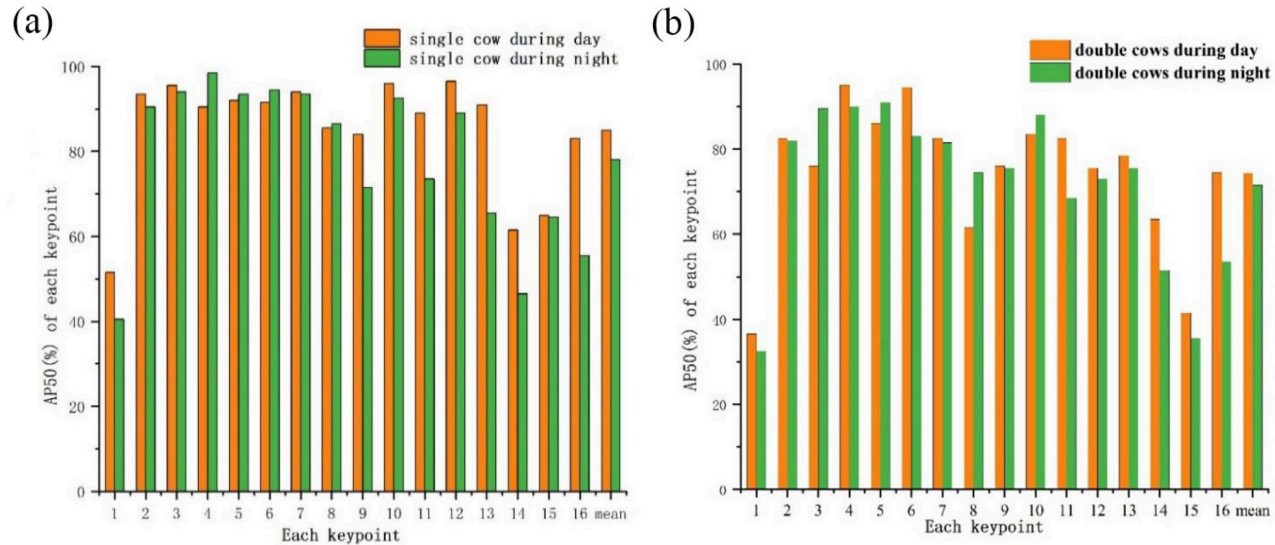

**Fig 6. Average precision of all keypoints.** (a) All keypoints of single cow. (b) All keypoints of double cows.

**Table 2. Confusion matrix of pose classification.**

| Actual poses | Predicted poses | | | |
|---|---|---|---|---|
| | Standing | Walking | Lying | Total |
| Standing | 121 | 9 | 0 | 130 |
| Walking | 10 | 119 | 1 | 130 |
| Lying | 1 | 0 | 129 | 130 |
| Total | 132 | 128 | 130 | 390 |
| Evaluation of multiclassification | | | | |
| Number of correct predictions | 121+119+129 = 369 | | | |
| Precision for each pose | 0.9167, 0.9297, 0.9923 | | | |
| Recall for each pose | 0.9308, 0.9154, 0.9923 | | | |
| Accuracy | 369/390 = 0.9462 | | | |

numbers on the main diagonal and the smaller numbers on other cells in the confusion matrix indicate better performance of the pose estimation algorithm. The confusion matrix of our pose estimation is listed in Table 2.

The first row in Table 2 shows that 121 standing poses were correctly predicted, and 9 were incorrectly predicted as walking. Similarly, 119 walking and 129 lying poses were correctly predicted in the second and third rows. The results showed that the number of correctly predicted poses was 369, and the number of incorrectly predicted poses was 21. The matrix of precision for each pose on the test set is illustrated in Fig 7. The precision vector was calculated as 0.9167, 0.9297, and 0.9923, and the recall vector was calculated as 0.9308, 0.9154, and 0.9923. Taking standing as an example, we can observe that the detection rates were 0.9167 for standing and 0.0758 for walking in all detected standing poses. The classification of standing and walking is relatively poor compared with lying. The characteristics of the two poses are very similar when the cow is standing or walking facing the camera. It is more difficult to distinguish between these two types of behaviors, which easily causes confusion. The average

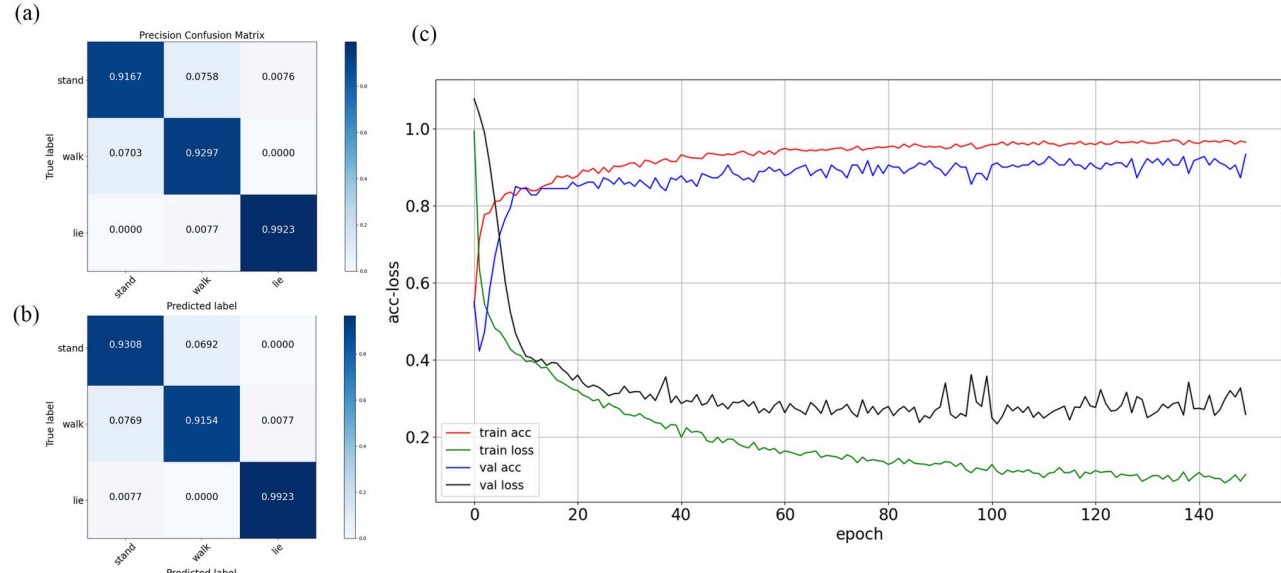

**Fig 7.** Matrix of (a) precision and (b) Recall ratio. (c) Average accuracy and loss function curves.

accuracy and loss function were calculated and compared for the training set and validation set. The accuracy and loss on the training set and validation set were 0.9648 and 0.1017 and 0.9278 and 0.2579, respectively.

## Discussion

In an actual open farm environment, various interference factors are present in the image obtained from the surveillance cameras, such as the varying light, occlusion between the cows and the representation of objects far away from the surveillance cameras. To verify the robustness of the algorithm and evaluate its effectiveness, we investigated the main factors that may affect pose estimation in the present study.

### Analysis of the influence of varying light on pose estimation

At the open cow breeding site, light in the early morning and twilight illuminates the surveillance camera. The images taken by the surveillance camera often encounter front-light, back-light or night conditions. The image is brighter under front light and darker under back light, which increases the difficulty of feature extraction. The low-level features of the convolutional layer contain more location and detailed information, and the high-level features have stronger semantic information. To reduce the impact of light on the features, we fused low-dimensional features such as texture, color, and edges with high-level cow semantic features to extract more image features. It is observed from the test results that the keypoint average precision of the multicow system at night was slightly lower than that during the day. The accuracy of the double cows situations was 74.3% during the day and 71.6% at night, showing only a small and not obvious difference. To simulate the impact of front-light and back-light farm environments without interference from other factors, we adjusted the brightness of the same input image by increasing or decreasing it by 25% and 50%. When the original image brightness was reduced by more than 50%, the surveillance camera activated the night vision function. To ensure that the test results were affected by unexpected factors rather than varying light, the same image was randomly selected from the test set. The test results are illustrated in Fig 8. The blue box is the predicted box of the detected cow, and the colored dots are the keypoints of the dairy cow' body. The pose label in blue font is displayed in the upper left corner of the predicted box. (a)

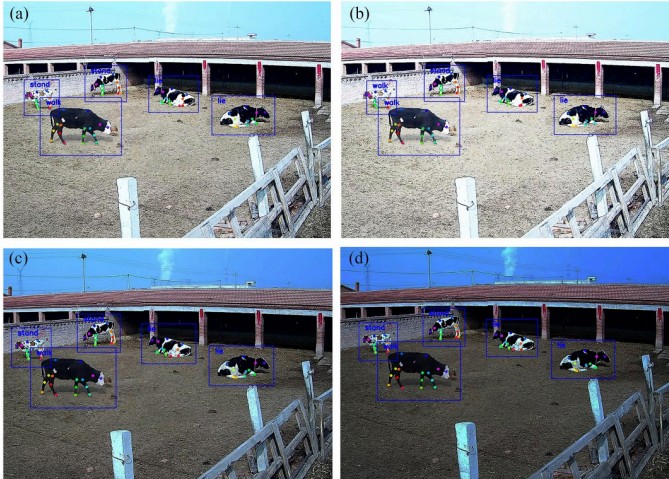

**Fig 8. The detection result at different brightness values.** (a) and (b) represent the influence of increasing the brightness by 25% and 50%. (c) and (d) represent the influence of decreasing the brightness by 25% and 50%.

and (b) show the influence of increasing the brightness by 25% and 50%. (c) and (d) show the influence of decreasing the brightness by 25% and 50%, respectively.

As seen from the above figure, the detection results of the same image containing 5 cows were basically the same after increasing or decreasing the brightness by 25% and 50%. As the brightness of the image changes, the three poses of the cow can still be accurately detected. The varying light had less effect on the accuracy of pose estimation in this study. The proposed algorithm of pose estimation under front lighting and backlighting exhibited high robustness.

## Analysis of the influence of occlusion on pose estimation

To verify the effectiveness of the algorithm under occlusion conditions, the test images were divided into two groups. Under the same light conditions, we compared the effects of different occlusion conditions on pose estimation. Among the test images, each category was collected under good lighting without occlusion, good lighting with occlusion, poor lighting without occlusion and poor lighting with occlusion.

As shown in Fig 9, the number of cows in the image ranged from single cow and double cows to 5 cows during the day, and the number also ranged from single cow, double cows to 6 cows at night. (a)-(c) show the pose estimation results under good lighting. (d)-(f) show the pose estimation results under poor lighting. There was obvious occlusion between two cows (front legs, head) and almost no occlusion between multiple cows. The results showed that the method proposed in this paper can still estimate the three typical poses with an increasing number of cows. An increase in the number of cows had less impact on the detection accuracy. The keypoint detection accuracy of a single cow was 85% during daylight and 78.1% at night, and that of double cows with occlusion was 74.3% and 71.6%. The mutual occlusion between cows reduced the keypoint detection accuracy under the same lighting conditions, resulting in the accuracy decline of pose classification. When the cow was occluded by more than 50%, the number of keypoints that could be detected was too small, so the accuracy of pose classification

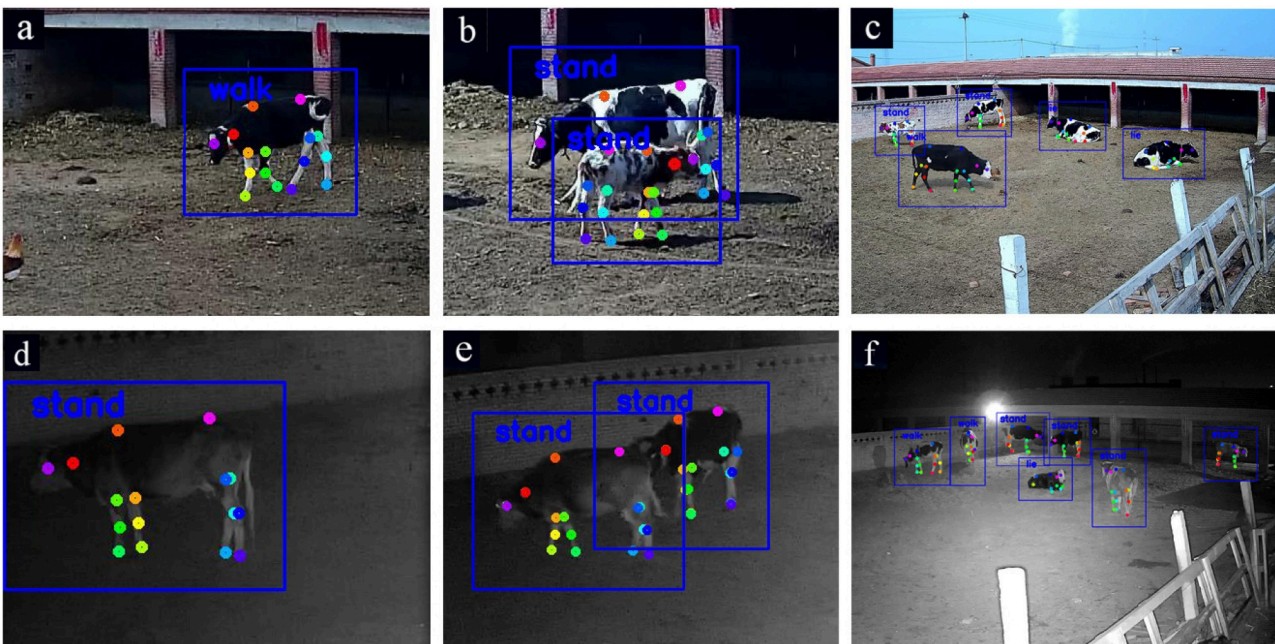

**Fig 9. Effects of different numbers of cows on pose estimation.** (a)-(c) indicate the image containing single cow, double cows and 5 cows during the day. (d), (e) and (f) represent the image containing single cow, double cows and 6 cows at night.

was relatively low. Nevertheless, cows are seldom seriously occluded in the actual cow sport field. The above experiments showed that the method proposed in this study exhibited good robustness for pose estimation under different occlusion conditions.

## Analysis of the influence of image resolution on precision

The shapes of cow images after the object detection network exhibited different sizes, and cows far from surveillance cameras appeared as small objects in the image, so that the resolution of small object images are relatively low. We compared the impacts of different resolutions of input images on the detection accuracy and speed in this study. Images with 300*168 pixels, 600*336 pixels and 900*504 pixels were selected to test keypoint detection in dairy cows. The average precision values were 25%, 52% and 77%, and the detection speeds were 4.15 fps, 3.69 fps, and 2.63 fps, respectively. Different image resolutions affected the accuracy and computational complexity of pose estimation. For the same network model, increasing the size of the input image improved the accuracy of pose estimation to a certain extent, but the computational cost was also greatly increased at the same time. Therefore, follow-up work will use the superresolution enhancement algorithm to change the image resolution to improve the detection accuracy of small objects far from the surveillance camera. We will reduce the number of parameter calculations by designing a lightweight network and improve the efficiency of the network without degrading its performance.

## Analysis of the influence of transitions between poses

As found by manual observation, cows become restless immediately during estrus or before calving. According to some previous research works [12, 13], specific irregular behaviors appear before calving and during estrus, such as lying, standing, frequently changing positions between lying down and standing up and crawling behavior. Since the network was trained based on standing, walking and lying images in this study, the failure detection rate was relatively high when the cows were changing poses (transitioning from lying to standing or transitioning from standing to lying). On the other hand, the pose characteristics of standing and walking are very similar, particularly when the cow's head is facing the camera. Standing and walking poses will be confused, and their classification precision is relatively poor compared with lying. As seen from the confusion matrix (Table 2), 7.58% of standing was incorrectly recognized as walking, and 7.03% of walking was recognized as standing. To overcome the abovementioned precision decrease, more data are needed to validate the findings of this research, particularly with respect to the transitions between poses.

## Conclusions

To extract three typical poses (standing, walking and lying) in an actual farm environment, we presented an algorithm for multiobject pose estimation based on transfer-learning methods. We analyzed the main influencing factors on the detection precision, such as the varying light, occlusion between cows and the analysis of small objects far away from the surveillance cameras.

The main conclusions are as follows.

1. In the cow breeding site, the collected images will appear with frontlight and backlight. To reduce the impact of light on features, we fused low-dimensional features such as texture, color, and edges with high-level cow semantic features to extract more image features. To simulate the farm environment of front light and back light without interference from other factors, we adjusted the same input image by increasing or decreasing the brightness

by 25% and 50%. The results showed that the change in natural light had little effect on the pose estimation algorithm proposed in this study.

2. To verify the effectiveness of the algorithm under occlusion conditions, we compared the influence of different occlusions under the same light conditions. The keypoint detection accuracy of a single cow was 85% in daylight and 78.1% at night, which was significantly higher than that for double cows with obvious occlusion (74.3% and 71.6%). The pose classification was also affected due to decline in keypoint accuracy resulting from occlusion. When the cow was occluded by more than 50%, the accuracy decreased significantly. In actual cow breeding site, cows are seldom seriously occluded. Therefore, the method of this study can be used for multicow pose estimation.

3. The shapes of cow images exhibited different sizes after the object detection network. Cows far away from the surveillance camera manifested as small objects in the image with relatively low resolution. We compared the pose detection results of three input images with different resolutions. As the image resolution improved, the detection accuracy also increased. Therefore, to improve the detection accuracy of small objects far from surveillance cameras, we will use the superresolution enhancement algorithm to change the image resolution in future work. We will reduce the number of parameter calculations by designing a lightweight network and improve the efficiency of the network without degrading its performance.

4. The failure detection rate was relatively high when the cows were changing poses between lying and standing and facing the camera. As found from the test results, 7.58% of standing was incorrectly recognized as walking, and 7.03% of walking was recognized as standing. To overcome the precision decrease, more data are needed to validate the findings of this research, particularly for transitions between poses.

The pose estimation and behavior classification methods of dairy cow based on skeleton feature extraction in this study have certain reference significance for animal behavior researchers. And this study can provide further data support for lameness detection, estrus detection and the prediction of calving in large-scale precision farming. However, the real-time performance of the algorithm was relatively poor. We will focus on improving the real-time performance while ensuring high detection accuracy and reducing the number of parameters and calculations in the following work.

## Author Contributions

**Conceptualization:** Caili Gong.

**Data curation:** Caili Gong, Xinyu Du, Lide Su.

**Formal analysis:** Caili Gong, Xinyu Du.

**Funding acquisition:** Yong Zhang, Yongfeng Wei, Zhi Weng.

**Investigation:** Caili Gong, Yongfeng Wei.

**Methodology:** Caili Gong.

**Project administration:** Yong Zhang.

**Resources:** Yongfeng Wei.

**Software:** Caili Gong, Xinyu Du.

**Supervision:** Yong Zhang.

**Validation:** Caili Gong, Xinyu Du.

**Visualization:** Caili Gong.

**Writing – original draft:** Caili Gong.

**Writing – review & editing:** Caili Gong, Yong Zhang, Zhi Weng.

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
