## [Decision Letter · Decision Letter 0]

21 Feb 2022

PONE-D-21-40261Multicows Pose Estimation Based on Keypoint ExtractionPLOS ONE

Dear Dr. Zhang,

Thank you for submitting your manuscript to PLOS ONE. After careful consideration, we feel that it has merit but does not fully meet PLOS ONE’s publication criteria as it currently stands. Therefore, we invite you to submit a revised version of the manuscript that addresses the points raised during the review process.

Please revise accordingly to the comments and suggestion by reviewer.

We look forward to receiving your revised manuscript.

Kind regards,

Yan Chai Hum

Academic Editor

PLOS ONE

Journal Requirements:

“NO, the funders had no role in study design, data collection and analysis, decision to publish, or preparation of the manuscript.

Reviewers' comments:

Reviewer's Responses to Questions

**Comments to the Author**

1. Is the manuscript technically sound, and do the data support the conclusions?

Reviewer #1: Yes

2. Has the statistical analysis been performed appropriately and rigorously? 

Reviewer #1: Yes

3. Have the authors made all data underlying the findings in their manuscript fully available?

Reviewer #1: No

4. Is the manuscript presented in an intelligible fashion and written in standard English?

Reviewer #1: Yes

5. Review Comments to the Author

Reviewer #1: The article presents an application of current detection methods to the poses of cows on a farm. The experiments have been explained and performed according to the field’s standard. A couple of points remain, that need attention before publication in my opinion.

1) Data availability

• additional information in the beginning of the PDF states, that all relevant data is freely available and that it is available within the manuscript and its Supporting Information files.

• But the manuscript does not contain any links to images, videos or code.

2) Ethics

• IACUC and approval number were not mentioned

◦ it was also not mentioned, that those were not necessary. Please state why the relevant authorities do not consider this an animal experiment, that needs to be approved. Or if they do, state the approval number. ◦

• How did non-contact machine vision improve animal welfare during recording?

◦ As stated under “data acquisition”

◦ perhaps you meant, that the research could in the future improve welfare?

3) Funding

• where was funding coming from? If the authors did not receive funding, it has to be stated

4) Data

• LabelImg was used, please give a source or citation

• Mosaic enhancement

◦ why did you use it? Why not other augmentation methods like mirroring or rotation?

◦ Should it not be called “mosaic augmentation” instead of “enhancement”

• How were the data set frames picked? It cannot be random from the videos, since you have exactly 600 images for each condition (standing, walking, lying). How exactly were validation and test set chosen?

• How often do the cows occlude each other by more than 50%? You state, that it is rather seldom in a real farm environment. Could you estimate how much it happens in your data?

5) Performance

• The abstract states, that the algorithm exhibited a higher detection rate. Higher than what?

• Table 1

◦ In the text and the table it says “IOU” in places, where in my understanding “IOU threshold” is meant. In case I misunderstand, please make it more clear, what “ average precision when the IOU is equal to 0.5” means. If it should indeed be “IOU threshold”, please correct.

◦ What is the difference between AP and AP50? They both appear in the rows, where the IOU threshold is 0.5. This confusion might be connected to the previous point.

◦ What are the “400 images tested simultaneously”?

6) Structure

• In the methods section the model setup is detailed. State more clearly what parts are done as in yolo v4 and what you have inserted yourself.

7) Conclusion

• It is necessary to assess at least roughly, whether the performance of the system would be adequate for applications in the area such as estrus detection. To make a statement under some assumptions in the conclusion would already be good. This would put the numbers into a useful context.

If these points are addressed I recommend publication after minor revision.

6. PLOS authors have the option to publish the peer review history of their article (what does this mean?). If published, this will include your full peer review and any attached files.

Reviewer #1: No

---

## [Author Response · Author response to Decision Letter 0]

23 Mar 2022

Dear Editor and Reviewer:

Thank you very much for your constructive comments and valuable suggestions. We have carefully considered the suggestion of Editor and Reviewer. We have tried our best to improve and made some changes in the manuscript according to your kind advices and detailed suggestions. The changes and corrections will be marked in red in the manuscript. This document summarizes our revisions and responses in the following. Thank you again for all your help with our manuscript.

Comments from Editors

Responds to the editors' comments:

Responds: Thank you very much for the kind reminder. We have downloaded the LaTeX template from PLOS ONE's official website and read the LaTeX guidelines. We have revised some of the formatting and will do our best to make our manuscript conform to PLOS ONE's style.

Responds: We are very sorry for the mismatch of grant information that was provided in the ‘Funding Information’ and ‘Financial Disclosure’ sections. We have revised the grant information and do our best to provide the correct grant numbers for the awards in the ‘Funding Information’ section.

“NO, the funders had no role in study design, data collection and analysis, decision to publish, or preparation of the manuscript.

Responds: We are very sorry for missing the funding information.We have listed the grants that supported our study in the ‘Funding Information’ section. This study was funded by the National Natural Science Foundation of China under Grant 61966026, Grant 62161034 and Grant 61561037. The funders had no role in study design, data collection and analysis, decision to publish, or preparation of the manuscript. The authors appreciate the funding organization for their financial supports. 

Responds: We are very sorry for our negligence of the links to images datasets. The minimal data set underlying the results described in our manuscript can be found at https://www.kaggle.com/twisdu/dairy-cow. All data in our manuscript will be fully shared without restriction. The link to the minimal dataset has been added in the Supporting Information files when submitting the revised manuscript.

Responds: Thank you very much for the kind reminder. The corresponding author have an ORCID iD and that it is validated in Editorial Manager. And we have updated the Information in PLOS ONE Editorial Manager Submission System.

Comments from Reviewers: 

Reviewer #1: The article presents an application of current detection methods to the poses of cows on a farm. The experiments have been explained and performed according to the field’s standard. A couple of points remain, that need attention before publication in my opinion.

Responds to the reviewer's comments:

1) Data availability

• additional information in the beginning of the PDF states, that all relevant data is freely available and that it is available within the manuscript and its Supporting Information files.

• But the manuscript does not contain any links to images, videos or code.

Responds: 

We are very sorry for our negligence of the links to images datasets. The minimal data set underlying the results described in our manuscript can be found at https://www.kaggle.com/twisdu/dairy-cow. All data in our manuscript will be fully shared without restriction. The link to the minimal dataset has been added in the Supporting Information files when submitting the revised manuscript.

2)Ethics

• IACUC and approval number were not mentioned

◦ it was also not mentioned, that those were not necessary. Please state why the relevant authorities do not consider this an animal experiment, that needs to be approved. Or if they do, state the approval number. 

• How did non-contact machine vision improve animal welfare during recording?

◦ As stated under “data acquisition”

◦ perhaps you meant, that the research could in the future improve welfare?

Responds: 

Thank you very much for your kindness suggestion. This study was carried out at Inner Mongolia Flag Animal Husbandry Co., Ltd. Inner Mongolia Agricultural University has conducted scientific research with Inner Mongolia Flag Animal Husbandry Co., Ltd. in Hohhot for more than five years. In the stage of experiment, the data was taken with the consent and direction of the company. The study does not require approval from any relevant authorities.

The traditional way based on manual observation methods to detect the behavior of dairy cows increases the probability of human-animal contact, which can lead to some zoonotic diseases. The way of wearable sensors to collect the behavior of dairy cows seriously interferes with the cows during the installation of the equipment, which is likely to cause a strong stress response to the dairy cows. The data in this paper was acquired by the monitoring camera, which was fixed on the fence of the breeding site at a height of 4 m. During the data collection process, neither the collection equipment nor the experimenter contacted the cows. It did not interfere with the normal activities of the cows, and did not cause the cows' stress response. Compared with traditional manual inspection and using wearable sensors, it can realize continuously monitoring the activity and health of dairy cows without contact. If the project could be implemented in the future, it can complete non-contact animal behavior monitoring and improve animal welfare. According to your kind suggestion, we have revised the detailed description of data acquisition in the manuscript

3) Funding

• where was funding coming from? If the authors did not receive funding, it has to be stated 

Responds: 

We are very sorry for missing the funding information. This research was funded by the National Natural Science Foundation of China under Grant 61966026, Grant 62161034 and Grant 61561037. The funders had no role in study design, data collection and analysis, decision to publish, or preparation of the manuscript. We have corrected the funding information in the ‘Funding Information’ section when submitting our revised manuscript.

4) Data

• LabelImg was used, please give a source or citation

• Mosaic enhancement

◦ why did you use it? Why not other augmentation methods like mirroring or rotation?

◦ Should it not be called “mosaic augmentation” instead of “enhancement”

• How were the data set frames picked? It cannot be random from the videos, since you have exactly 600 images for each condition (standing, walking, lying). How exactly were validation and test set chosen?

• How often do the cows occlude each other by more than 50%? You state, that it is rather seldom in a real farm environment. Could you estimate how much it happens in your data?

Responds: 

Thank you very much for your detailed suggestion. 

LabelImg: The source of LabelImg annotation tool is https://github.com/tzutalin/labelImg. We have added the source citation in the revised manuscript.

Mosaic augmentation: As your kind suggestion, we revised the expression to Mosaic augmentation. It is a kind of data augmentation method. It has the advantage of enriching the background of the detection content. The data of 4 images can be calculated at a time during Batch Normalization (BN) calculation. The 4 images were mirrored, flipped, cropped before they were stitched together.

Data set: The data frames were manually selected so that the selected images contain each condition (standing, walking, lying). The validation and test sets are randomly drawn from the overall dataset.

Occlusion: During the experiment, we adjusted the height of the camera to make the cow objects less occluded in video. To test the robustness of cow occlusion effects on detection accuracy, we lowered the shooting height in some video. When the little cow body was occluded by more than 50%, the detection accuracy significantly decreased. As your kind suggestion, we will continue to study the occlusion situation to improve the detection accuracy and meet the needs of large-scale precision farming.

5) Performance

• The abstract states, that the algorithm exhibited a higher detection rate. Higher than what?

• Table 1

◦ In the text and the table it says “IOU” in places, where in my understanding “IOU threshold” is meant. In case I misunderstand, please make it more clear, what “ average precision when the IOU is equal to 0.5” means. If it should indeed be “IOU threshold”, please correct.

◦ What is the difference between AP and AP50? They both appear in the rows, where the IOU threshold is 0.5. This confusion might be connected to the previous point.

◦ What are the “400 images tested simultaneously”?

Responds: 

Thank you very much for your constructive comments and valuable suggestions. As your kind suggestion, we revised the expression in manuscript. This study combined YOLO v4 object detection and pose estimation to classify daily behaviors of multi-cow. The Test result showed that the detection accuracy was higher than that in previous research results. Song et al. [6] proposed a skeleton extraction model of cows in walking states with a high accuracy rate of up to 93.40% when the OKS was 0.75. Hahn Klimroth et al. [22] presented a multistep convolutional neural network for detecting three typical poses of African ungulates, obtaining a high accuracy of 93%. Chen et al. [24] proposed an algorithm based on YOLACT with high detection speed and accuracy for real-time detection and tracking of multiple parts of pig bodies. The detect accuracy of the algorithm in the data set could reach up to 90%,

In Table 1, IoU means the Intersection over Union, and IoU threshold is a judgment threshold. If the IoU of the predicted box and the ground truth is greater than or equal to the IoU threshold, the predicted result is considered to be TP; otherwise, the predicted result is considered to be FP. When IoU is equal to 0, the prediction result is considered to be FN. AP50 is the average precision when the IoU threshold is set to 0.5. AP is the mean value when the value of IOU is taken from 0.5 to 0.95 and the step size is 0.05.

The 400 images refer to the image test set containing all four cases listed in the table.

6) Structure

• In the methods section the model setup is detailed. State more clearly what parts are done as in yolo v4 and what you have inserted yourself.

Responds: 

Thank you very much for your constructive comments and valuable suggestions. Considering the Reviewer’s suggestion, we have added the detailed statement of the improvement YOLO v4. Firstly, the original loss function consists of three parts, namely the bounding box regression loss, the confidence loss and the category loss. In this paper, we only study on the cow objects, category loss does not need to be considered. We modified the overall loss function of the object detection network into confidence loss and regression loss. Confidence loss was used to describe whether there was an object in the grid cell by calculating the binary cross entropy. Regression loss was used to describe the position and size difference between the annotated object and the predict object. Then, we adopted the transfer learning in this paper. We used the pretrained network weights on the PASCAL VOC 2012 dataset and initialized the network model. And then the network was fine-tuned by self-built dataset to achieve better training effect. We have revised the presentation of this section in the manuscript.

7) Conclusion

• It is necessary to assess at least roughly, whether the performance of the system would be adequate for applications in the area such as estrus detection. To make a statement under some assumptions in the conclusion would already be good. This would put the numbers into a useful context.

Responds:

Thank you very much for your valuable suggestions. When a cow is in lameness, estrus and before calving, the pose will change frequently. The pose estimation and behavior classification methods of dairy cow based on skeleton feature extraction in this study have certain reference significance for animal behavior researchers. And this study can provide further data support for lameness detection, estrus detection and the prediction of calving in large-scale precision farming. We will focus on improving the real-time performance while ensuring high detection accuracy and reducing the number of parameters and calculations in the following work.

---

## [Editor Report · Decision Letter 1]

12 Apr 2022

PONE-D-21-40261R1Multicow Pose Estimation Based on Keypoint ExtractionPLOS ONE

Dear Dr. Zhang,

Thank you for submitting your manuscript to PLOS ONE. After careful consideration, we feel that it has merit but does not fully meet PLOS ONE’s publication criteria as it currently stands. Therefore, we invite you to submit a revised version of the manuscript that addresses the points raised during the review process.

Please improve each of the figure's caption and table's title such that these captions or titles are self-contained in which the take-home message of the figure/table can be described as part of the caption so that the purpose of the figure or table can be easily delivered to the ready, even without referring to the texts. In your "response to comments" for your revised article, please detail the improvement for each of the figure's caption and table's title (before and after) and explain how these captions or titles have been improved. If you wish to remain the current caption or table's title, please explain the reasons for consideration.

We look forward to receiving your revised manuscript.

Kind regards,

Yan Chai Hum

Academic Editor

PLOS ONE

Journal Requirements:

Additional Editor Comments:

Please revise the figures captions and table titles so that they can standalone or self-contained such that the main message of the figure/table is narrated as part of the caption.
---

## [Author Response · Author response to Decision Letter 1]

6 May 2022

Dear Editor:

Thank you very much for your constructive comments and valuable suggestions. We have carefully considered the suggestion of Editors and Reviewers. We have tried our best to improve and made some changes in the manuscript according to your kind advices and detailed suggestions. We added the ethics statement at the beginning of the Materials and methods section of our manuscript file. The changes and corrections will be marked in red in the manuscript. This document summarizes our revisions and responses in the following. Thank you again for all your help with our manuscript. We sincerely hope this manuscript will be finally acceptable to published on PLOS ONE. 

Comments from Editor

Responds to the editor's comments:

1.Please improve each of the figure's caption and table's title such that these captions or titles are self-contained in which the take-home message of the figure/table can be described as part of the caption so that the purpose of the figure or table can be easily delivered to the ready, even without referring to the texts. In your "response to comments" for your revised article, please detail the improvement for each of the figure's caption and table's title (before and after) and explain how these captions or titles have been improved. If you wish to remain the current caption or table's title, please explain the reasons for consideration.

Responds: 

Thank you very much for the kind suggestion. We have revised each of the figure's caption and table's title and will do our best to make our manuscript conform to PLOS ONE's style.

Fig.1: As your kind suggestion, we revised the caption from ‘Keypoint label template of cow body’ to ‘Keypoints label template of the cow skeleton’. In this study, we estimated the cow skeleton through 16 keypoints and the partial affinity field between these keypoints. It showed the labeled positions and order of the 16 skeleton keypoints in Fig.1. The caption “Keypoints label template of the cow skeleton” can accurately describe the meaning of the current figure.

Fig.2: The figure’s caption before was “Overview of the pose analysis procedure.” The revised figure’s caption are: (a) Overview of pose analysis procedure. (b) Close-up of the multiobject detector based on YOLOv4. Fig.2(a) is the overall process of the entire pose analysis, and Fig.2(b) present a detailed description of the object detector based YOLO v4. We added the detailed description of the two figures, so that the figure could be easily understood, even without referring to the texts. 

Fig.3: Thank you very much for your kindness suggestion. We are very sorry for the inaccurate figure’s description in original manuscript. As your kind suggestion, we revised the caption from “Close-up of keypoint extraction model” to “The structure of two-branch skeleton extraction network”. The network structure in Fig.3 consists of two branches: the upper branch is responsible for predicting the location of keypoints, and the lower branch is responsible for the partial affinity field between keypoints. As a result, the new description are able to more accurately represent the meaning of the figure. 

Fig.4: We retain the description of Figure 4 “Fully connected neural network of pose classification.” After extracting the skeleton information of the cow’s body in the image, a fully connected neural network was designed to classify three typical behaviors. We think that the classification network structure can be described by “Fully connected neural network of pose classification”.

Fig.5: Thank you very much for your kindness suggestion. The figure’s caption before was “The OKS values of the keypoints in different scenarios”. The revised figure’s caption is “The confidence of various keypoints in different scenarios”. The OKS value is the confidence parameter for keypoint detection. If only the OKS value is used to describe figure 5, it is not easy to understand without the textual representation of the paper. As your kind suggestion, we revised the caption to “The confidence of various keypoints in different scenarios”. 

Fig.6: We are very sorry for the lack of detailed figure’s description in original manuscript. The figure’s caption before was “AP values of all keypoints. The revised figure’s caption: Average precision of all keypoints. (a) All keypoints of single cow. (b) All keypoints of double cows. We added the detailed description of the two figures, so that the figure could be easily understood, even without referring to the texts. 

Fig.7: Thank you very much for your kindness suggestion. We are very sorry for the lack of detailed figure’s description in original manuscript. The figure’s caption before: (a) Precision Matrix. (b) Average accuracy and loss function curves. The revised figure’s caption: Matrix of (a) precision and (b) Recall ratio. (c)Average accuracy and loss function curves. The description of recall ratio was missing in the original description of figure 7(b), we have improved the detailed caption in the revised manuscript.

Fig.8: We are very sorry for the lack of detailed figure’s description in original manuscript. The figure’s caption before:The influence of varying light on pose estimation. The revised figure’s caption: The detection result at different brightness values. (a) and (b) represent the influence of increasing the brightness by 25% and 50%. (c) and (d) represent the influence of decreasing the brightness by 25% and 50%. The lighting changes for each figure were not explicitly stated in the previous representation. As your kind suggestion, we have added more detailed descriptions for each figure, so that the figure could be easily understood, even without referring to the texts. 

Fig.9: The figure’s caption before: Pose estimation results under different scenarios. The revised figure’s caption: Effects of different numbers of cows on pose estimation. (a)-(c) indicate the image containing single cow, double cows and 5 cows during the day. (d), (e) and (f) represent the image containing single cow, double cows and 6 cows at night. We are very sorry for the lack of detailed figure’s description in original manuscript. We have added detailed descriptions for each figure in the revised version. 

Table 1: The table's title before: Detection results under different scenarios. The revised figure’s caption: Evaluation indicators of object detection under different scenarios. Table 1 shows several parameters that measure the results of object detection, such as precision, recall, average precision (AP), AP50 and AP75. These parameters are the evaluation indicators of object detection. Therefore, we believe that “Evaluation indicators of object detection under different scenarios” can be more accurately describe the Table 1.

Table 2: The table's title before: Detection results under different scenarios 

The revised figure’s caption: Confusion matrix of pose classification. We are very sorry for the inaccurate table's title in original manuscript. In most multiclassification tasks, the evaluation metrics are computed based on the

confusion matrix. In table 2, the confusion matrix was also used as the evaluation metric of three typical pose classifications. 

Responds: 

Thank you very much for the kind reminder. We have rechecked the reference list to ensure that it is complete and correct. In the revised manuscript, the reference list has not been changed.

3.Please upload a Response to Reviewers letter which should include a point by point response to each of the points made by the Editor and / or Reviewers. (This should be uploaded as a 'sponse to Reviewers'ile type.) Please follow this link for more information: http://blogs.PLOS.org/everyone/2011/05/10/how-to-submit-your-revised-manuscript/

Responds: 

Thank you very much for the kind reminder. We uploaded a Response to Reviewers letter which included a point by point response to each of the points made by the Editor and Reviewers. 

4.Thank you for including your ethics statement on the online submission form: "This study was carried out at Inner Mongolia Flag Animal Husbandry Co., Ltd. Inner Mongolia Agricultural University has conducted scientific research with Inner Mongolia Flag Animal Husbandry Co., Ltd. in Hohhot for more than five years. In the stage of experiment, the data was taken with the company'approved. The study does not require approval from the relevant authorities. The data was acquired by the monitoring camera, which was fixed on the fence of the breeding site at a height of 4 m. During the data collection process, neither the collection equipment nor the experimenter contacted the cows. It did not interfere with the normal activities of the cows, and did not cause the cows'tress response. Compared with traditional manual inspection and using wearable sensors, it can realize continuously monitoring the activity and health of dairy cows without contact. If the project could be implemented in the future, it can complete non-contact animal behavior monitoring and improve animal welfare." To help ensure that the wording of your manuscript is suitable for publication, would you please also add this statement at the beginning of the Methods section of your manuscript file.

Responds: 

As your kind suggestion, we have added an ethical statement at the beginning of the materials and methods section of the revised manuscript, marked in red font.

Comments from Reviewers: 

Reviewer #1: The article presents an application of current detection methods to the poses of cows on a farm. The experiments have been explained and performed according to the field’s standard. A couple of points remain, that need attention before publication in my opinion.

Responds to the reviewer's comments:

1) Data availability

• additional information in the beginning of the PDF states, that all relevant data is freely available and that it is available within the manuscript and its Supporting Information files.

• But the manuscript does not contain any links to images, videos or code.

Responds: 

We are very sorry for our negligence of the links to images datasets. The minimal data set underlying the results described in our manuscript can be found at https://www.kaggle.com/twisdu/dairy-cow. All data in our manuscript will be fully shared without restriction. The link to the minimal dataset has been added in the Supporting Information files when submitting the revised manuscript.

2)Ethics

• IACUC and approval number were not mentioned

◦ it was also not mentioned, that those were not necessary. Please state why the relevant authorities do not consider this an animal experiment, that needs to be approved. Or if they do, state the approval number. 

• How did non-contact machine vision improve animal welfare during recording?

◦ As stated under “data acquisition”

◦ perhaps you meant, that the research could in the future improve welfare?

Responds: 

Thank you very much for your kindness suggestion. This study was carried out at Inner Mongolia Flag Animal Husbandry Co., Ltd. Inner Mongolia Agricultural University has conducted scientific research with Inner Mongolia Flag Animal Husbandry Co., Ltd. in Hohhot for more than five years. In the stage of experiment, the data was taken with the consent and direction of the company. The study does not require approval from any relevant authorities.

The traditional way based on manual observation methods to detect the behavior of dairy cows increases the probability of human-animal contact, which can lead to some zoonotic diseases. The way of wearable sensors to collect the behavior of dairy cows seriously interferes with the cows during the installation of the equipment, which is likely to cause a strong stress response to the dairy cows. The data in this paper was acquired by the monitoring camera, which was fixed on the fence of the breeding site at a height of 4 m. During the data collection process, neither the collection equipment nor the experimenter contacted the cows. It did not interfere with the normal activities of the cows, and did not cause the cows' stress response. Compared with traditional manual inspection and using wearable sensors, it can realize continuously monitoring the activity and health of dairy cows without contact. If the project could be implemented in the future, it can complete non-contact animal behavior monitoring and improve animal welfare. According to your kind suggestion, we have revised the detailed description of data acquisition in the manuscript

3) Funding

• where was funding coming from? If the authors did not receive funding, it has to be stated 

Responds: 

We are very sorry for missing the funding information. This research was funded by the National Natural Science Foundation of China under Grant 61966026, Grant 62161034 and Grant 61561037. The funders had no role in study design, data collection and analysis, decision to publish, or preparation of the manuscript. We have corrected the funding information in the ‘Funding Information’ section when submitting our revised manuscript.

4) Data

• LabelImg was used, please give a source or citation

• Mosaic enhancement

◦ why did you use it? Why not other augmentation methods like mirroring or rotation?

◦ Should it not be called “mosaic augmentation” instead of “enhancement”

• How were the data set frames picked? It cannot be random from the videos, since you have exactly 600 images for each condition (standing, walking, lying). How exactly were validation and test set chosen?

• How often do the cows occlude each other by more than 50%? You state, that it is rather seldom in a real farm environment. Could you estimate how much it happens in your data?

Responds: 

Thank you very much for your detailed suggestion. 

LabelImg: The source of LabelImg annotation tool is https://github.com/tzutalin/labelImg. We have added the source citation in the revised manuscript.

Mosaic augmentation: As your kind suggestion, we revised the expression to Mosaic augmentation. It is a kind of data augmentation method. It has the advantage of enriching the background of the detection content. The data of 4 images can be calculated at a time during Batch Normalization (BN) calculation. The 4 images were mirrored, flipped, cropped before they were stitched together.

Data set: The data frames were manually selected so that the selected images contain each condition (standing, walking, lying). The validation and test sets are randomly drawn from the overall dataset.

Occlusion: During the experiment, we adjusted the height of the camera to make the cow objects less occluded in video. To test the robustness of cow occlusion effects on detection accuracy, we lowered the shooting height in some video. When the little cow body was occluded by more than 50%, the detection accuracy significantly decreased. As your kind suggestion, we will continue to study the occlusion situation to improve the detection accuracy and meet the needs of large-scale precision farming.

5) Performance

• The abstract states, that the algorithm exhibited a higher detection rate. Higher than what?

• Table 1

◦ In the text and the table it says “IOU” in places, where in my understanding “IOU threshold” is meant. In case I misunderstand, please make it more clear, what “ average precision when the IOU is equal to 0.5” means. If it should indeed be “IOU threshold”, please correct.

◦ What is the difference between AP and AP50? They both appear in the rows, where the IOU threshold is 0.5. This confusion might be connected to the previous point.

◦ What are the “400 images tested simultaneously”?

Responds: 

Thank you very much for your constructive comments and valuable suggestions. As your kind suggestion, we revised the expression in manuscript. This study combined YOLO v4 object detection and pose estimation to classify daily behaviors of multi-cow. The Test result showed that the detection accuracy was higher than that in previous research results. Song et al. [6] proposed a skeleton extraction model of cows in walking states with a high accuracy rate of up to 93.40% when the OKS was 0.75. Hahn Klimroth et al. [22] presented a multistep convolutional neural network for detecting three typical poses of African ungulates, obtaining a high accuracy of 93%. Chen et al. [24] proposed an algorithm based on YOLACT with high detection speed and accuracy for real-time detection and tracking of multiple parts of pig bodies. The detect accuracy of the algorithm in the data set could reach up to 90%,

In Table 1, IoU means the Intersection over Union, and IoU threshold is a judgment threshold. If the IoU of the predicted box and the ground truth is greater than or equal to the IoU threshold, the predicted result is considered to be TP; otherwise, the predicted result is considered to be FP. When IoU is equal to 0, the prediction result is considered to be FN. AP50 is the average precision when the IoU threshold is set to 0.5. AP is the mean value when the value of IOU is taken from 0.5 to 0.95 and the step size is 0.05.

The 400 images refer to the image test set containing all four cases listed in the table.

6) Structure

• In the methods section the model setup is detailed. State more clearly what parts are done as in yolo v4 and what you have inserted yourself.

Responds: 

Thank you very much for your constructive comments and valuable suggestions. Considering the Reviewer’s suggestion, we have added the detailed statement of the improvement YOLO v4. Firstly, the original loss function consists of three parts, namely the bounding box regression loss, the confidence loss and the category loss. In this paper, we only study on the cow objects, category loss does not need to be considered. We modified the overall loss function of the object detection network into confidence loss and regression loss. Confidence loss was used to describe whether there was an object in the grid cell by calculating the binary cross entropy. Regression loss was used to describe the position and size difference between the annotated object and the predict object. Then, we adopted the transfer learning in this paper. We used the pretrained network weights on the PASCAL VOC 2012 dataset and initialized the network model. And then the network was fine-tuned by self-built dataset to achieve better training effect. We have revised the presentation of this section in the manuscript.

7) Conclusion

• It is necessary to assess at least roughly, whether the performance of the system would be adequate for applications in the area such as estrus detection. To make a statement under some assumptions in the conclusion would already be good. This would put the numbers into a useful context.

Responds:

Thank you very much for your valuable suggestions. When a cow is in lameness, estrus and before calving, the pose will change frequently. The pose estimation and behavior classification methods of dairy cow based on skeleton feature extraction in this study have certain reference significance for animal behavior researchers. And this study can provide further data support for lameness detection, estrus detection and the prediction of calving in large-scale precision farming. We will focus on improving the real-time performance while ensuring high detection accuracy and reducing the number of parameters and calculations in the following work.

---

## [Editor Report · Decision Letter 2]

18 May 2022

Multicow Pose Estimation Based on Keypoint Extraction

PONE-D-21-40261R2

Dear Dr. Zhang,

We’re pleased to inform you that your manuscript has been judged scientifically suitable for publication and will be formally accepted for publication once it meets all outstanding technical requirements.

Kind regards,

Yan Chai Hum

Academic Editor

PLOS ONE

Additional Editor Comments (optional):

all concerns have been addressed.
---

## [Editor Report · Acceptance letter]

23 May 2022

PONE-D-21-40261R2 

Multicow Pose Estimation Based on Keypoint Extraction 

Dear Dr. Zhang:

I'm pleased to inform you that your manuscript has been deemed suitable for publication in PLOS ONE. Congratulations! Your manuscript is now with our production department. 

Kind regards, 

on behalf of

Dr. Yan Chai Hum 

Academic Editor

PLOS ONE